# Analysis of Signal Processing Methods to Reject the DC Offset Contribution of Static Reflectors in FMCW Radar-Based Vital Signs Monitoring

**DOI:** 10.3390/s22249697

**Published:** 2022-12-10

**Authors:** Marco Mercuri, Tom Torfs, Maxim Rykunov, Stefano Laureti, Marco Ricci, Felice Crupi

**Affiliations:** 1Dipartimento di Informatica, Modellistica, Elettronica e Sistemistica (DIMES), University of Calabria, 87036 Rende, Italy; 2Imec, 3001 Leuven, Belgium

**Keywords:** DC offset calibration, Doppler, FMCW, heart rate, mmWave, phase demodulation, remote radar sensing, respiration rate, sub-10 GHz radar, vital signs monitoring

## Abstract

Frequency-modulated continuous wave (FMCW) radars are currently being investigated for remote vital signs monitoring (measure of respiration and heart rates) as an innovative wireless solution for healthcare and ambient assisted living. However, static reflectors (furniture, objects, stationary body parts, etc.) within the range or range angular bin where the subject is present contribute in the Doppler signal to a direct current (DC) offset. The latter is added to the person’s information, containing also a useful DC component, causing signal distortion and hence reducing the accuracy in measuring the vital sign parameters. Removing the sole contribution of the unwanted DC offset is fundamental to perform proper phase demodulation, so that accurate vital signs monitoring can be achieved. In this work, we analyzed different DC offset calibration methods to determine which one achieves the highest accuracy in measuring the physiological parameters as the transmitting frequency varies. More precisely, by using two FMCW radars, operating below 10 GHz and at millimeter wave (mmWave), we applied four DC offset calibration methods to the baseband radar signals originated by the cardiopulmonary activities. We experimentally determined the accuracy of the methods by measuring the respiration and the heart rates of different subjects in an office setting. It was found that the linear demodulation outperforms the other methods if operating below 10 GHz while the geometric fitting provides the best results at mmWave.

## 1. Introduction

In the last two decades, frequency-modulated continuous wave (FMCW) radar sensors have been extensively investigated for remote vital signs monitoring, namely for contactless measure of the respiration and the heart rates of a subject [1,2,3]. This opens a multitude of healthcare and ambient assisted living applications, especially when wearable medical devices cannot be used (e.g., on patients with severe and extensive burn wounds), create discomfort, and are unpleasant for long term use (e.g., while sleeping) [4,5].

One of the biggest challenges of accurate remote vital signs monitoring performance is to distinguish the very weak contribution of the physiological movements from the disturbances caused by the surrounding environment. The FMCW radar has the capability of separating the electromagnetic reflections in time. This allows dividing the monitoring environment into range bins, using a single-input single-output (SISO) architecture, or in range angular bins using multiple-input multiple-output (MIMO) and beam steering architectures. In case of the former, the range fast Fourier transform (range-FFT) processing is used on the radar data, while the range-FFT processing and the beamforming operation are performed for the latter [6,7]. The vital signs monitoring is performed by extracting the Doppler signal from the range or range angular bin where the cardiopulmonary activity is sensed, i.e., locating the subject’s thorax. The latter operation can be performed (1) by knowing the position a priori, hence selecting the corresponding bin; (2) by determining the highest variation in the range spectrum [6]; (3) by applying array signal processing techniques [7]. The capability of dividing the monitoring environment in range or range angular bins allows the effect of static reflectors (e.g., clutter, objects, furniture, stationary body part, etc.) to be reduced, though this still remains a problem to deal with. In fact, since the radar has finite range and angular resolutions, inevitable static reflectors within the range or range angular bin where the subject is present contribute in the Doppler signal to an overall direct current (DC) offset. The latter is added to the person’s information (which containing also a useful DC component) significantly distorting the phase signal related to the vital signs, thus jeopardizing the monitoring [8,9]. DC offset calibration becomes hence a pivotal preprocessing operation before performing a proper phase demodulation [8,9]. The goal is to identify the contribution due to the vital signs and to separate it from the disturbance, i.e., the DC offset, caused by the static reflectors within the range or range angular bin.

In this work, we analyzed four DC offset calibration methods using two FMCW radars operating within the two most used worldwide ultra-wideband (UWB) unlicensed frequency ranges for indoor applications, namely 7.25–8.5 GHz and 57–64 GHz at millimeter wave (mmWave) [5,6,7,8,9,10,11,12,13,14,15,16]. The goal was to experimentally determine which technique achieves the highest accuracy in measuring the vital sign rates in function of the transmitting frequency. To our best knowledge, this is the first time that such a comparison has been performed involving an UWB architecture at different frequencies. In fact, two similar studies have been performed in the past only by using a 2.4 GHz CW radar for vital signs [17] and structural health monitoring [18]. However, current trend in this field is to employ UWB systems that offer several advantages over the 2.4 GHz CW radar such as multi-people tracking (localization and speed information), concurrent localization and vital signs monitoring on multiple subjects, capability of providing both angular and range information, reducing (but not eliminating) the effect of static reflectors by leveraging the range/angular resolution [18,19,20,21,22,23,24,25,26,27,28,29,30,31,32]. We specify that the findings obtained with the 2.4 GHz CW radar in [17,18] are not directly applicable at the frequencies considered in this work. This is because (1) the lower the frequency, the lower the phase (Doppler) excursion due to the vital signs; (2) operating at a single frequency involves oscillators with better phase noise and phase error performances; (3) the higher the frequency, the lower the signal-to-noise ratio (SNR) of the extracted baseband signals. Those three points strongly influence the selection of the DC calibration method as the frequency varies.

## 2. Methods

### 2.1. Radar Signal Modelling

The cardiopulmonary activity, namely the physiological movements of the heart and lungs of a person, causes sub-millimeter motions on the skin surface of the thoracic area. Due to the Doppler effect, the phase shift caused by these motions and embedded into the reflected signal can be detected by a radar, so that the respiration and the heart rates can be measured [33]. The complex baseband signal *B*(*t*), obtained after mixing the received signal with a copy of the transmitted signal, can be modelled as
(1)$${B(t)=A(t)\;e}^{j\phi \left(t\right)}{=A(t)\;e}^{j\frac{4\pi y(t)}{\lambda }},$$
where *A*(*t*) is the signal amplitude, *ϕ*(*t*) is the Doppler information, *λ* is the wavelength referred to the first frequency of the FMCW signal (*chirp*), and *y*(*t*) is the chest motion due to the cardiopulmonary activity. The latter is the vital signs information to be extracted, and it can be approximated as
(2)$${\;y(t)=Y}_{b}\cos\left({2\pi f}_{b}t\right){+Y}_{h}\cos\left({2\pi f}_{h}t\right),$$
where *Y_b_* and *Y_h_* are the sub-millimeter amplitudes of the movements caused by the lungs and heart on the chest surface (with typical amplitudes of 4–12 mm and 0.1–0.5 mm, respectively), and *f_b_* and *f_h_* are the breathing and the heart rates, respectively [11]. We refer the reader to [13] for a detailed spectral analysis of Equation (1).

By determining the angular information of Equation (1), it is possible to retrieve *y*(*t*) from which we can extract the vital sign rates. It must be noted that static reflectors (i.e., furniture, objects, etc.) as well as the stationary body parts within the range or range angular bin where the subject is present contribute to an overall signal of magnitude *A_s_* and phase *θ_s_*. This is real case scenario in daily life situation. Equation (1) can be then rewritten as
(3)$${B(t)=A}_{s}{\;e}^{{j\theta }_{s}}{+A(t)\;e}^{j\frac{4\pi y(t)}{\lambda }},\;$$
where the first term is the DC offset and the second one is subject’s information. The latter consists of a variable part and of DC information that is fundamental to perform a linear phase demodulation [9]. The goal of the DC offset calibration methods is to eliminate only the contribution of the DC offset (first term) while fairly preserving the subject’s information, namely the DC information and the variable part. Thus, the main challenge is to distinguish the DC information from the DC offset.

In practical circumstance, *A_s_* is stronger than *A*(*t*). Under such a condition, Equation (3) can be approximated as [34]
(4)$$\;B(t)\;\approx {\;A}_{T}{(t)\;e}^{j\left\{\theta_{s}+\frac{A(t)\;}{A_{s}}\sin\left[\theta_{s}-\;\frac{4\pi y(t)}{\lambda }\right]\right\}},\;$$
where *A_T_*(*t*) is the overall magnitude. We can note that *y*(*t*) is non-linearly combined in the exponential argument and it is also multiplied by *A*(*t*)/*A_s_* << 1. In this situation, the resulting Doppler (phase) signal is distorted, and the vital signs information *y*(*t*) cannot be accurately retrieved.

Figure 1 shows a graphical illustration in the complex plane of Equations (3) and (4) where I (In-phase) and Q (Quadrature) are its real and imaginary components, respectively. The cardiopulmonary activity involves a rotating vector which describes in the complex plane a small arc (red solid line) when working with sub-10 GHz radars or a circle (red dashed line) when operating at mmWave. The static reflectors contribute to an overall DC offset, pushing the arc/circle away from the origin (magenta vector). This results in a non-linear combination of θ*_s_* and *ϕ*(*t*), as modelled in Equation (4), jeopardizing the extraction of the vital signs information. For a proper phase demodulation, and hence to accurately extract the vital signs information *y*(*t*), the arc/circle must be centered to the origin of the complex plane, hence performing a DC calibration. This centering operation eliminates the effect of the static reflectors (i.e., the magenta vector), leaving only the contribution of the cardiopulmonary activity as modelled in Equation (1).

### 2.2. AC-Coupling

The simplest DC calibration technique is to perform alternate current (AC) coupling operation directly to Equation (3). Assuming to arrange *B*(*t*) into a matrix **x** = [**x_1_**, **x_2_**] with **x_1_** and **x_2_** being, respectively, the I and Q components, this method consists in removing the mean values of **x_1_** and **x_2_**. The phase demodulation is performed extracting the angular information of *B*(*t*) as
(5)$$\;\arctan\left[\frac{x_{2\;}-\;\mathrm{mean}(x_{2})}{x_{1\;}-\;\mathrm{mean}(x_{1})}\right].\;$$

### 2.3. Linear Demodulation

This method can be used under the small angle condition, hence when 4*πy*(*t*)/*λ* << 1 in Equation (3). In such a situation, the trajectory of the baseband signal in the complex plane is a small arc, meaning that this technique can be used only for sub-10 GHz radars.

The idea behind the linear demodulation is to rotate and to move the arc parallel to the Q-axis. The result is a demodulated signal being the projection of the arc to the Q-axis. This means that the phase variation, i.e., the arc, is approximated as a segment. Therefore, the smaller the angular variation, the higher the accuracy in phase demodulating the signal. The steps for performing the linear demodulation are:

1.Removing the mean values of **x_1_** and **x_2_** and combining into a matrix $\bar{x}$ = [**x_1_**-mean(**x_1_**), **x_2_**-mean(**x_2_**)];

2.Calculating the covariance matrix of $\bar{x}$;

3.Obtaining the matrix **E** whose columns are the eigenvectors of the covariance matrix;

4.Multiplying the transpose of **E** by $\bar{x}$. The result is a matrix containing the principal components listed in descending order depending on the eigenvalues. The first principal component is the demodulated signal.

### 2.4. Minimizing the Algebraic Distance

This technique estimates a circle in the complex plane from the I and Q signals. It can be used both for sub-10 GHz and for mmWave radar. In the latter case, although the IQ components already describe a circle in the complex plane, this method is still necessary to reduce the effect of the noise and outliers.

The general form of the equation of a circle is
(6)$${a\;x}^{T}x+b\;x+c=0,\;$$
where *a* ≠ 0, **b** = (*b*_1_, *b*_2_) and *c* are the coefficients to be computed to fit the circle from **x**. Equation (6) can be rewritten as [35]
(7)$${\left(x_{1}+\frac{b_{1}}{2a}\right)}^{2}+{\left(x_{2}+\frac{b_{2}}{2a}\right)}^{2}=\frac{{∥b∥}^{2}}{{4a}^{2}}-\frac{c}{a},\;$$
from which the center coordinates **z** = (*z*_1_, *z*_2_
) and the radius can be respectively determined as
(8)$$z=\left(z_{1}{,z}_{2}\right)=\left(-\frac{b_{1}}{2a},-\frac{b_{2}}{2a}\right)\;$$
and
(9)$$\;r=\sqrt{\frac{{∥b∥}^{2}}{{4a}^{2}}-\frac{c}{a}}.\;$$

Hence, the phase demodulation can be performed computing the angular information after subtracting the center points from the IQ signals as
(10)$$\;\phi (t)=\frac{4\pi y(t)}{\lambda }=\arctan\left(\frac{x_{2\;}{-\;z}_{2}}{x_{1\;}{-\;z}_{1}}\right).$$

### 2.5. Minimizing the Geometric Distance

With this method, the center coordinates **z** are estimated by solving a nonlinear least-squares geometric fitting problem [35]. The objective function is
(11)$$\underset{z,r}{\min}\overset{N}{\underset{n=1}{\sum }}d_{n}{\left(z,r\right)}^{2},\;$$
where *n* = 1, …, *N*, with *N* being the number of measured points, and
(12)$$d_{n\;}^{2}=\;\;{\left(∥z\;-\;x\left(n\right)∥\;-\;r\right)}^{2},\;$$
which represents the geometric distance between the *n*-th data point and the circle. The algebraic solution is a good starting vector for methods minimizing the geometric distance. The angular information can be then determined using Equation (10).

## 3. Material

### 3.1. Radar Sensors

We used two different FMCW radar sensors, whose parameters are listed in Table 1. One device is the sub-10 GHz imec Mercurius V1.1 radar sensor [11]. The other device is the commercial Texas Instruments IWR6843ISK mmWave radar sensor.

### 3.2. Reference Sensor

The NeXus-10 MkII device (FDA approved and CE class IIa conform), integrating an electrocardiograms (ECG) sensor and respiration belt, has been used to provide reference measurements for the respiration and the heart rates. The physiological signals have been acquired with a sampling rate of 256 Hz.

## 4. Experimental Validation

### 4.1. Signal Processing for Vital Sign Extraction and Data Collection

For each measurement, we extracted the complex baseband signal from the bin corresponding to the location of the subject’s thorax. The localization is performed by determining the highest variation in the range spectrum (this approach was also used in [6]). We lowpass filtered the complex baseband signals before applying the DC offset calibration methods and performing phase demodulation to extract the vital sign (Doppler) signals. We retrieved the respiration and heartbeat signals after filtering the Doppler signals and estimated the corresponding rates through FFT. Note that both radar devices operated in a single-channel mixer mode and that the resulting IQ signals have been obtained after performing the FFT-based range processing. This ensures a perfect quadrature between the two signals.

We conducted the experimental validation in a typical office setting with 12 subjects, 10 males and 2 females, differing in height (155–195 cm), weight, and age (20–45 years). Only a single volunteer at a time was present in the room, which contained furniture to increase the effect of the static reflectors. Each subject was normally breathing, sitting on a chair at 1.5 m away from the radar, which was fixed at 1.2 m above the floor. A total of 24 measurements of 2 min have been collected. More precisely, we first collected half of the measurements with the sub-10 GHz imec Mercurius V1.1 radar and then the other half with the mmWave radar. Each measurement was processed by applying all the four methods described in Section 2 and considering sliding windows of 30 s with overlaps of 5 s. It should be specified that the linear demodulation was not applied to the data acquired with the mmWave radar since at those operating frequencies the small angle condition is no longer valid.

### 4.2. Results

In this Section, we present the results of the experimental validation. We determined the mean absolute errors (MAEs) and the root mean square errors (RMSEs) expressed in terms of BPM which stands, respectively, for breaths per minute and beats per minute when it refers to respiration rate (RR) and heart rate (HR).

Figure 2 and Figure 3 show the boxplots obtained, respectively, with the sub-10 GHz and mmWave radars.

In Figure 4 and Figure 5, we presented the results using the Bland–Altmann plots obtained, respectively, with the sub-10 GHz and mmWave radars. With the latter, for both RR and HR estimation, the best performing method resulted in an average bias very close to 0 (Figure 5c,d). In contrast, the worst method, while still having an average bias of around 0.4 BPM for the RR, presents an average bias close to 3 BPM for the HR estimation, indicating an overestimation on average. The confidence intervals are also higher in the case of the worst method. Furthermore, a systematic error can be appreciated by looking at Figure 5b,d. Regardless of the method used, the HR estimated with the radar seems to be overestimating higher HR and underestimating lower HR. This could be due to the fact that measuring low HRs might involve errors since the HR fundamental get closer to the RR harmonics (especially the 2nd–3rd). Moreover, small random body motion can influence the HR measurements, especially above 1 Hz. However, in general, sub-10 GHz radars involve higher SNR than mmWave radars.

Finally, we report the average MAEs and RMSEs in Table 2 and Table 3.

## 5. Discussion

### 5.1. Sub-10 GHz Radar

As it is reported in Table 2 and shown in Figure 2, the linear demodulation outperforms the other methods in retrieving the physiological parameters. However, the geometric fitting technique achieves similar and satisfactory results. We will justify and explain those findings with the help of Figure 6 and Figure 7.

Figure 6 shows the baseband signals in the IQ plane before and after applying the DC calibration methods. As expected, they describe small arcs. Figure 6a shows both the original arc, i.e., before applying any DC calibration method, and the one after AC coupling. A yellow circle centered to the origin and intersecting the original arc is also shown. Although the original arc lies on the yellow circle, it can be seen that it is not perfectly aligned with it, hence it is not centered properly to the origin of the complex plane. This is due to the combined effect of static reflectors and stationary body parts. In the same way, the other arc is not centered to the origin but lies on it, as the AC coupling also removes the subject’s DC information. Figure 6b shows the arc after applying the linear demodulation. Like the AC coupling method, the arc is not properly centered to the origin. However, the key difference is that the arc is properly aligned with the Q-axis. By exploiting the small angle approximation, the arc is well approximated to its projection onto the axis. Finally, Figure 6c,d show the arc after minimizing the algebraic and geometric distances, respectively. We have also indicated in orange the fitted circles obtained with the two methods. It is possible to notice that only the geometric fitting yields a proper DC calibration. Although minimizing the algebraic distance results in a simple method, this fit is often unsatisfactory for small arcs. However, the algebraic solution can be used as a starting vector for minimizing the geometric distance.

Figure 7 compares the vital sign spectra obtained with the reference device and after applying the DC offset calibration methods to the baseband signals of Figure 6. More precisely, Figure 7a refers to the respiration rate while Figure 7b to the heart rate. In this example, only the outputs of the linear demodulation and geometric fitting methods are aligned with the references. This is because the AC coupling does not center the arc to the origin while the circle is not always properly estimated with the algebraic fitting, resulting in an improper DC offset calibration.

Although the geometric fitting yields satisfactory results, in this validation, the linear demodulation turned out to be the best method. This can be explained by the fact that the estimation of the circle is strongly dependent on the nominal position of the subject, hence any (even small) random motions can move the circle to another point of the plane. Therefore, also the length of the window signal influences the circle fitting: the longer the window, the higher the accuracy. In the same way, the longer the window, the higher the probability of random motions. As opposite to that, the linear demodulation does not estimate a circle and its results are more robust in presence of outliers (e.g., baselines in the Doppler shift, random body movements).

### 5.2. mmWave Radar

For processing with the mmWave radar, we have not considered the linear demodulation as the small angle approximation condition is no longer valid. As shown in Table 2, both the algebraic and geometric fitting methods achieve satisfactory results, although the latter results are significantly better in measuring the heart rate. As in Section 5.1, we use two figures to argue the findings. In an ideal situation at mmWave, the rotating vector corresponding to the baseband signals describes a circle of several radians in the IQ plane. However, in practice, the vector describes a figure which resembles a partial spiral (Figure 8) due to noise and random body (thoracic) motion. We will refer to that figure as a circle hereinafter.

Figure 8 shows the original circle and the ones obtained after applying the DC calibration techniques. In Figure 8a, we can notice that the original circle is not perfectly centered to the origin as it presents a small and unwanted DC offset. In an ideal situation, i.e., with only the contribution of the thoracic motion, the circle would be centered to the origin. Figure 8b–d shows the results after applying the DC offset calibration methods. It is hard to notice significant differences with the naked eye. However, although the centering operations produce slightly different outputs (see Figure 9), they have a significant impact in accurately retrieving the vital signs parameters. In Figure 8c,d, we have plotted also the fitted circle.

Figure 9a,b show, respectively, the respiration rate and the heart rate obtained from the signal of Figure 8. In addition, in this case, we have compared them with the relative references. As we can see both from Table 3 and Figure 9a, all the three methods result accurate in retrieving the respiration rates. At mmWave, the thoracic motion due to the lungs produce a phase excursion of several radians. Therefore, even an imperfect DC calibration yields satisfactory results. However, this is not valid when retrieving the heartbeat which involves a phase excursion of a fraction of π. In this case, the AC coupling results are inaccurate with respect to those achieved using the algebraic and geometric fitting methods.

### 5.3. Comparison with the State-of-the-Art

In Table 4, we compared the best results of this work with the ones reported in some reference works based on UWB architectures.

## 6. Conclusions

In this paper, we analyzed DC offset calibration methods for remote radar-based respiration and heart rates monitoring at two different operating frequencies. We conclude that the linear demodulation outperforms the other methods if operating at sub-10 GHz frequencies. In fact, we reported MAEs of 0.32 BPM and 1.19 BPM and RMSEs of 0.70 BPM and 2.07 BPM, respectively, for the respiration rate and for the heart rate as a result of the experimental validation performed. On the other hand, minimizing the geometric distance provides the best results at mmWave. In this respect, we reported MAEs of 0.22 BPM and 2.72 BPM and RMSEs of 0.45 BPM and 3.65 BPM, respectively, for the respiration rate and for the heart rate. The same considerations can be drawn from the boxplots and from the Bland–Altman plots presented in Section 4.2. Finally, it should be noted that the geometric fitting works well despite the frequency used.

## Figures and Tables

**Figure 1 sensors-22-09697-f001:**
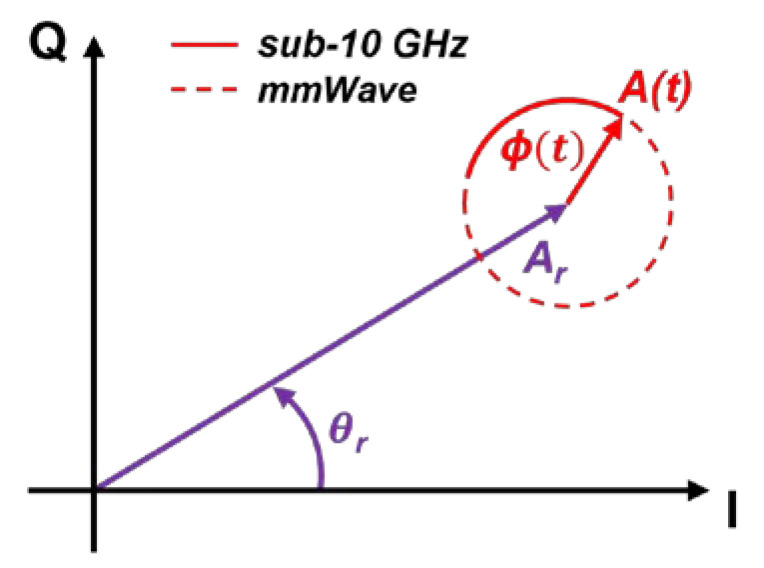
Graphical illustration in the complex plane of the baseband signal resulting from the cardiopulmonary activity and static reflectors.

**Figure 2 sensors-22-09697-f002:**
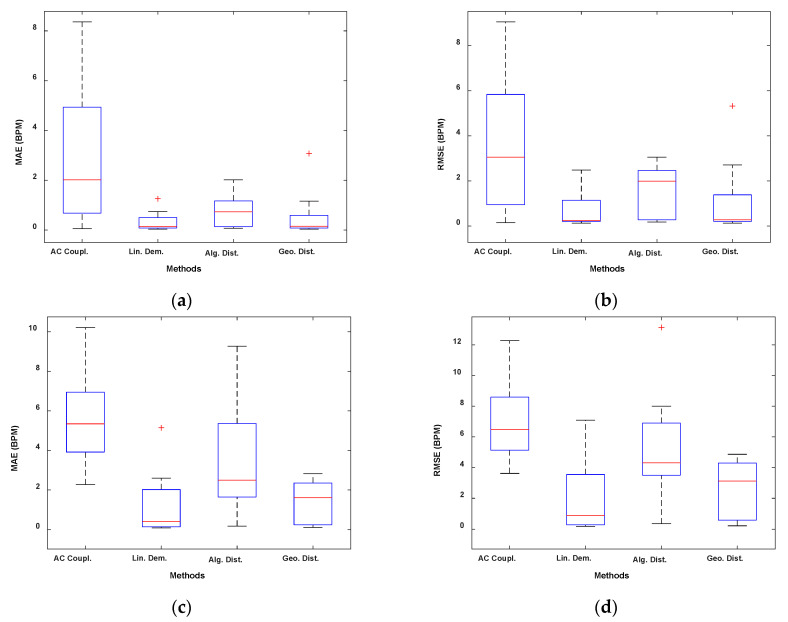
Boxplots of the experimental validation using the sub-10 GHz radar. (**a**) MAE RR. (**b**) RMSE RR. (**c**) MAE HR. (**d**) RMSE HR.

**Figure 3 sensors-22-09697-f003:**
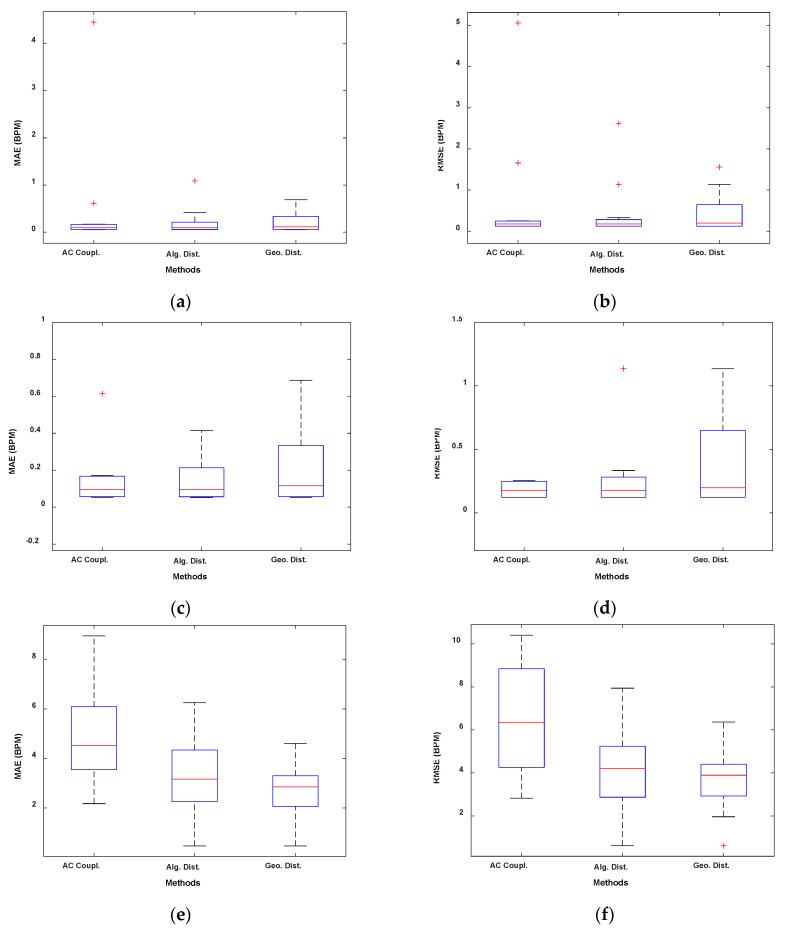
Boxplots of the experimental validation using the mmWave radar. (**a**) MAE RR. (**b**) RMSE RR. (**c**) Zoomed version of (**a**). (**d**) Zoomed version of (**b**). (**e**) MAE HR. (**f**) RMSE HR.

**Figure 4 sensors-22-09697-f004:**
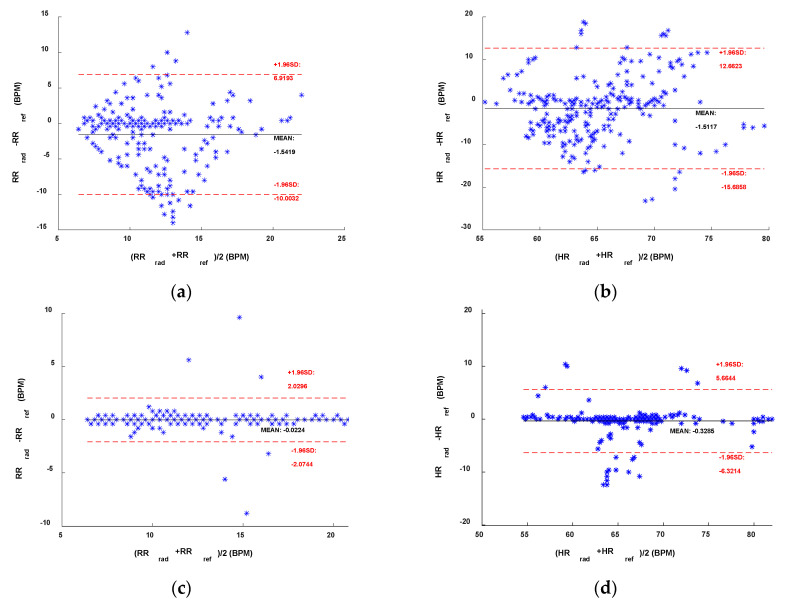
Bland–Altman comparisons of the references and the sub-10 GHz radar. (**a**) RR and **(b**) HR with the AC Coupling method. (**c**) RR and (**d**) HR with the linear demodulation method.

**Figure 5 sensors-22-09697-f005:**
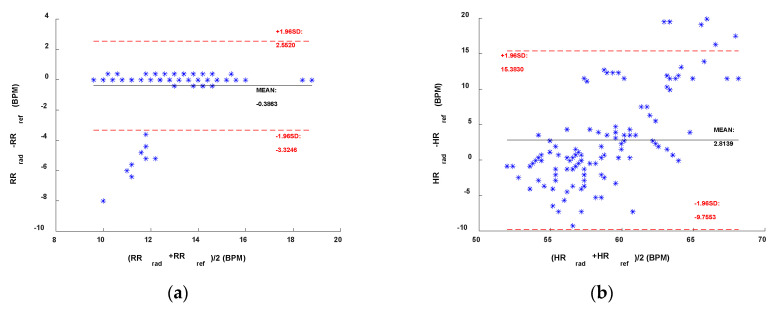
Bland–Altman comparisons of the references and the mmWave radar. (**a**) RR and (**b**) HR with the AC Coupling method. (**c**) RR and (**d**) HR with the geometric fitting method.

**Figure 6 sensors-22-09697-f006:**
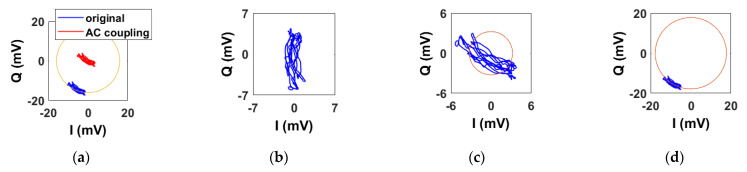
Example of baseband signals in the IQ plane obtained with the sub-10 GHz radar after applying the DC calibration methods. (**a**) AC coupling. (**b**) Linear demodulation. (**c**) Algebraic distance. (**d**) Geometric distance.

**Figure 7 sensors-22-09697-f007:**
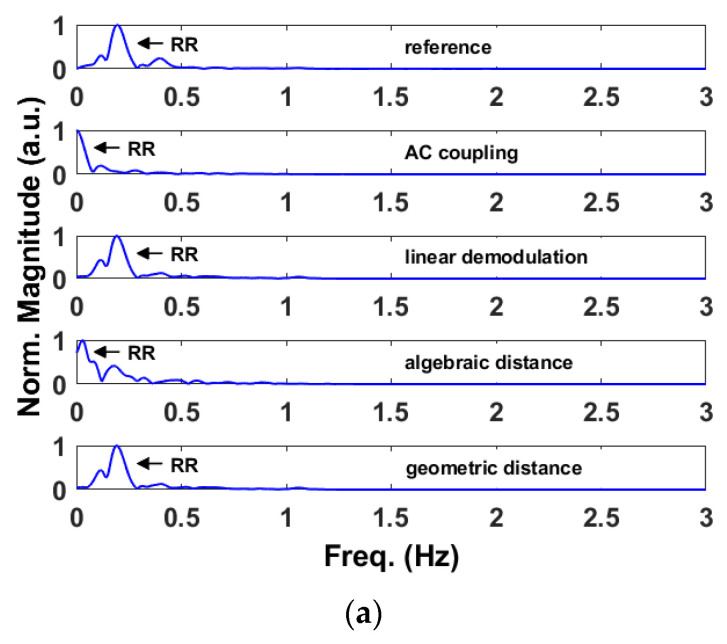
Example of baseband signals in the IQ plane obtained with the sub-10 GHz radar after applying the DC calibration methods. (**a**) Spectra obtained from the retrieved respiration signals. (**b**) Spectra obtained from the retrieved heartbeat signals.

**Figure 8 sensors-22-09697-f008:**
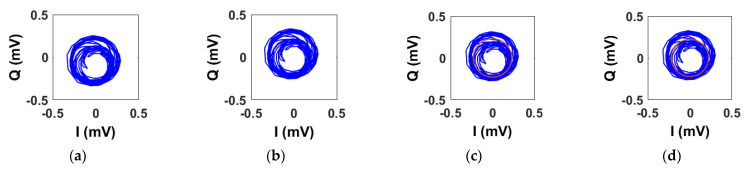
Example of baseband signals in the IQ plane obtained with the mmWave radar after applying the DC calibration methods. (**a**) Original signal. (**b**) AC coupling. (**c**) Algebraic distance. (**d**) Geometric distance.

**Figure 9 sensors-22-09697-f009:**
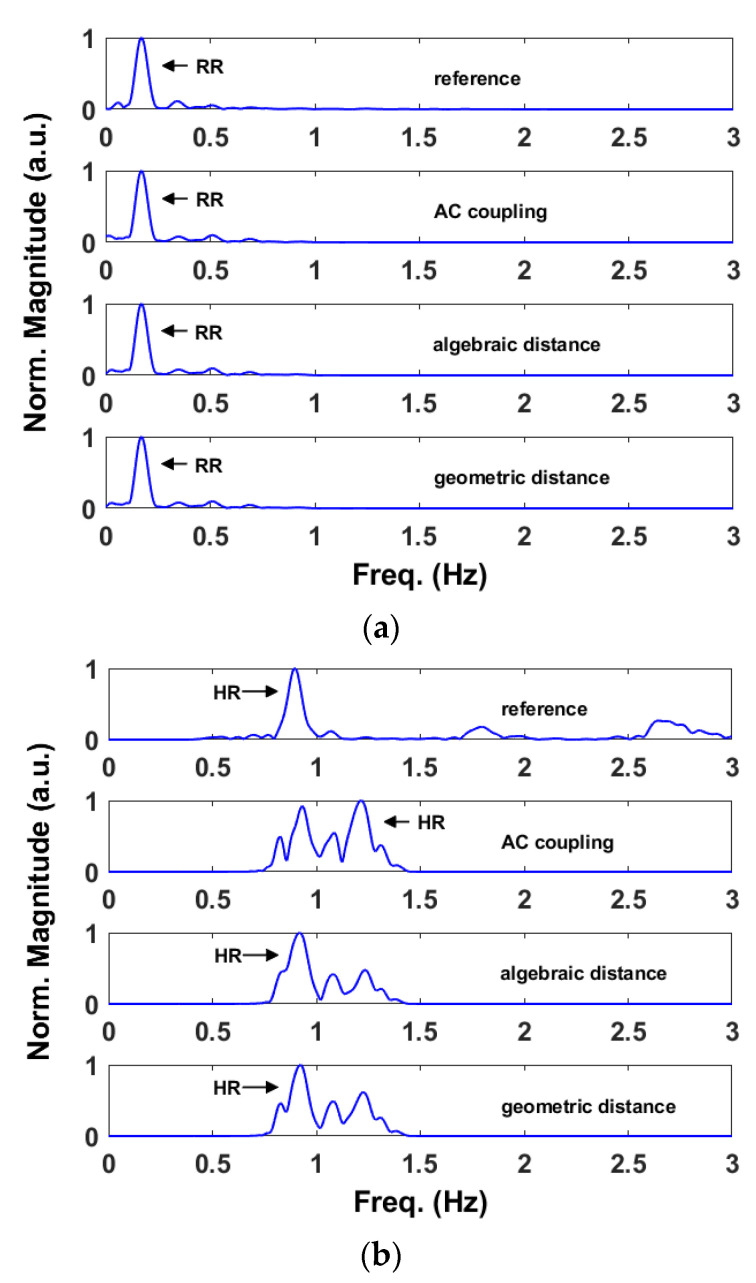
Example of baseband signals in the IQ plane obtained with the mmWave radar after applying the DC calibration methods. (**a**) Spectra obtained from the retrieved respiration signals. (**b**) Spectra obtained from the retrieved heartbeat signals.

**Table 1 sensors-22-09697-t001:** Radar parameters.

Parameters	Imec Mercurius V1.1	IWR6843ISK
Starting Frequency	7.3 GHz	60.645 GHz
Total Bandwidth	750 MHz	3.7 GHz
Chirp Duration	102.4 µs	64 µs
Fast Time Sampling Rate	10 MHz	4 MHz
Slow Time Sampling Rate	325.52 Hz	20 Hz
Range Resolution	20 cm	4 cm

**Table 2 sensors-22-09697-t002:** Experimental results using the sub-10 GHz radar.

Errors	AC Coupling	Linear Demodulation	Algebraic Distance	Geometric Distance
MAE RR (BPM)	2.90	0.32	0.77	0.56
RMSE RR (BPM)	3.62	0.70	1.63	1.12
MAE HR (BPM)	5.46	1.19	3.36	1.41
RMSE HR (BPM)	6.97	2.07	5.06	2.55

**Table 3 sensors-22-09697-t003:** Experimental results using the mmWave radar.

Errors	AC Coupling	Algebraic Distance	Geometric Distance
MAE RR (BPM)	0.50	0.21	0.22
RMSE RR (BPM)	0.70	0.46	0.45
MAE HR (BPM)	4.87	3.23	2.72
RMSE HR (BPM)	6.56	4.04	3.65

**Table 4 sensors-22-09697-t004:** Comparison table.

Reference	Central Freq. (GHz)	MAE RR (BPM)	RMSE RR (BPM)	MAE HR (BPM)	RMSE HR (BPM)
[5]	8.5	2.3	1.8	n.a.	n.a.
[20]	5.8	0.8	n.a.	3.1	n.a.
[36]	2.5	0.71	0.72	1.04	1.11
[37]	9.6	n.a.	n.a.	5.15	n.a.
This work	7.675	0.32	0.70	1.19	2.07
[3]	61.945	1.04	n.a.	3.71	n.a.
[14]	60	0.19	n.a.	0.92	n.a.
[22]	60	n.a.	n.a.	2.26	3.26
[25]	77	n.a.	0.66	n.a.	3.60
This work	62.495	0.22	0.45	2.72	3.65

## Data Availability

The data presented in this study are available on request from the corresponding author. The data are not publicly available due to privacy issues.

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
