# Peer review of "Analysis of Signal Processing Methods to Reject the DC Offset Contribution of Static Reflectors in FMCW Radar-Based Vital Signs Monitoring"

_sensors, 2022, doi:10.3390/s22249697_

Round 1

Reviewer 1 Report

Good work that interests researchers in this specialty. The text flow is good and coherent. The results are presented clearly and extensively. A compact conclusion is practical as well.

Author Response

We thank the Reviewer for their kind revision and for appreciating our work.

Reviewer 2 Report

In this work, the authors compare different methods to remove unwanted DC components from radar signals used for vital sign estimation. The paper is well written and easy to understand. To improve the paper significantly, I suggest that the authors amend their results section. We only learn about the results of their methods in two tables (2 & 3) presenting only mean values, we don’t learn anything about the distribution of the error or the distribution of the reference heart rates / respiratory rates.

For one, I suggest the authors additionally present their results as boxplots, where every sample consists of the RMSE for one subject. Thus, the reader gets an intuition whether the differences are significant.

For another, I suggest the authors include Bland-Altmann-Plots, maybe one for the best method and one for the worst method. Then we get to know how much the ground truth rates actually vary.

Minor comments:

Why do you use “heart beat”? Why not call it respiratory rate and heart rate?

Line 101 “details on how (1)…” – do you mean indeed (1) or (2) here?

In my opinion, the figures can be made smaller to safe some space.

Author Response

In this work, the authors compare different methods to remove unwanted DC components from radar signals used for vital sign estimation. The paper is well written and easy to understand. To improve the paper significantly, I suggest that the authors amend their results section. We only learn about the results of their methods in two tables (2 & 3) presenting only mean values, we don’t learn anything about the distribution of the error or the distribution of the reference heart rates / respiratory rates.

We thank the Reviewer for their kind revision and comments/suggestions which helped to improve the quality of the manuscript.

For one, I suggest the authors additionally present their results as boxplots, where every sample consists of the RMSE for one subject. Thus, the reader gets an intuition whether the differences are significant.

We added the boxplots in Figs. 2 and 3.

For another, I suggest the authors include Bland-Altmann-Plots, maybe one for the best method and one for the worst method. Then we get to know how much the ground truth rates actually vary.

We added the Bland-Altmann plots in Figs. 4 and 5.

Minor comments:

Why do you use “heart beat”? Why not call it respiratory rate and heart rate?

As suggested by the Reviewer, we have now used the term heart rate both in the text and in the figures.

Line 101 “details on how (1)…” – do you mean indeed (1) or (2) here?

Yes, we refer to equation (1). To avoid confusion we now wrote: “We refer the reader to [13] for a detailed spectral analysis of Eq. (1).”

In my opinion, the figures can be made smaller to safe some space.

We modified the figures’ sizes in the revised manuscript as suggested by the Reviewer.

Reviewer 3 Report

- Do you have ethical approval to perform measurements on humans?

- Page 6, line 213:  Tables II and III not Tables I and II.

- Page 7, line 223:  Tables II  not  Tables I .

- Page 7, line 239, Figures 2c,d not Figures 3c,d 

- Page 9: Figure 3. Example of baseband signals in the IQ plane obtained with the sub-10 GHz radar after applying the DC calibration methods. (a) AC coupling. (b) Linear demodulation. (c) Algebraic distance. (d) Geometric distance: Where are figures 3.c and 3.d?

- Page 9, line 275: As in Subsection IV.B, there is no subsection IV.B, please correct.

- Page 11, Where are figures 5.c and 3.d?

- Page 10, line 297, Table II or III inn this case?

- In the discussion paragraph, there is no comparison with further works or papers

- Conclusion paragraph can be improved

Author Response

We thank the Reviewer for their kind revision and comments which helped to improve the manuscript.

- Do you have ethical approval to perform measurements on humans?

As stated in the manuscript, the study was conducted in accordance with the Declaration of Helsinki, and approved by the IMEC ethical board (protocol ID: IP-19-WATS-TIP2-056). We provided to the Editorial Board the ethical approval and the informed consent forms.

- Page 6, line 213:  Tables II and III not Tables I and II.

Indeed, they are Table 2 and Table 3. We corrected it.

- Page 7, line 223:  Tables II  not  Tables I .

It is indeed Table 1. We corrected it.

- Page 7, line 239, Figures 2c,d not Figures 3c,d

We corrected it.

- Page 9: Figure 3. Example of baseband signals in the IQ plane obtained with the sub-10 GHz radar after applying the DC calibration methods. (a) AC coupling. (b) Linear demodulation. (c) Algebraic distance. (d) Geometric distance: Where are figures 3.c and 3.d?

There was indeed a mistake in the caption. We corrected with: “Figure 7. Example of baseband signals in the IQ plane obtained with the sub-10 GHz radar after applying the DC calibration methods. (a) Spectra obtained from the retrieved respiration signals. (b) Spectra obtained from the retrieved heartbeat signals.

- Page 9, line 275: As in Subsection IV.B, there is no subsection IV.B, please correct.

We corrected it.

- Page 11, Where are figures 5.c and 3.d?

There was indeed a mistake in the caption. We corrected with: “Figure 9. Example of baseband signals in the IQ plane obtained with the mmWave radar after applying the DC calibration methods. (a) Spectra obtained from the retrieved respiration signals. (b) Spectra obtained from the retrieved heartbeat signals.

- Page 10, line 297, Table II or III inn this case?

Indeed, it is Table 3. We corrected it.

- In the discussion paragraph, there is no comparison with further works or papers

We added Subsection 5.3 and Table 4. We compared the results of this works with the ones obtained in some significant manuscripts published in literature.

- Conclusion paragraph can be improved

We improved the conclusion paragraph adding the values obtained from the experimental validation.

Round 2

Reviewer 2 Report

The authors have adressed most of my comments in a satisfactorial manner and the manuscript was improved, thanks a lot. There are two remaining points.

I suggest to change the upper y-limits in Figure 3 a & b to 1.5 BMP and mention in the caption that individual outliers are capped OR add another figure with these limits so that the reader can get a better idea on how the boxes look like.

I suggest a more critical evaluation of the Plots 5 b and 5 d: For one, the limits of agreement are close to the distribution of the ground truth. It should also be stated that the BA plot shows that there is a systematic error in the HR estimation (noticable by the rotated error distribution). Also: are the authors sure the BA plots are calculated correctly? It seems that high HR values are overestimated and low HR values are underestimated (we often see the oposite behaviour which could indicate that the y-axis is flipped).

Author Response

We added a pdf file containing point-by-point response to the Reviewer’s comments.

Reviewer 3 Report

I am satisfied with the corrections made.

Best regards

Author Response

(The authors gave the same response as above.)
